# Integrated Reservoir Model and Differential Stimulation Modes of Low Permeability Porous Carbonate Reservoir: A Case Study of S Reservoir in X Oilfield in Iraq

**Jing Yang [1],\*, Guangya Zhu [1], Yichen An [2], Nan Li [1], Wei Xu [1], Li Wan [3] and Rongrong Jin [1]**

[1] Research Institute of Petroleum Exploration & Development, PetroChina, Beijing 100083, China; zhuguangya69@petrochina.com.cn (G.Z.); linan2017@petrochina.com.cn (N.L.); xu_wei@petrochina.com.cn (W.X.); jrrjrr@petrochina.com.cn (R.J.)

[2] Halfaya Oil Co., PetroChina Halfaya Company, Beijing 100034, China; anyichen@petrochina-hfy.com

[3] China Petroleum Engineering, China National Petroleum Corporation, Beijing 100101, China; wanli@petrochina-hfy.com

\* Correspondence: yangjing299@petrochina.com.cn

**Abstract:** The S reservoir in the X Oilfield in Iraq has great development potential due to its rich geological reserves. However, the low permeability and strong heterogeneity of the reservoir lead to great differences in reservoir stimulation performance. In this study, an integrated reservoir model and differential stimulation mode are put forward to solve the above problems. First, the feasibility of fracturing is evaluated by laboratory experiments. Second, an integrated reservoir model is established, which mainly includes a rock mechanics model, fracturing simulation model, and numerical simulation model, and correct the integrated model by fracturing operation curves and production dynamic curves. Third, three types of stimulation areas are classified according to the combination of sweet spot types, and three different stimulation modes are proposed. In conclusion, a small-scale stimulation mode should be applied in the Type I area to maximize economic benefits. In the Type II area, the medium-scale stimulation mode should be performed to ensure certain productivity while achieving certain economic benefits. In the Type III area, the large-scale stimulation mode should be employed to obtain certain productivity while economic benefits must be above a limit. The differential stimulation model proposed in this paper has made a great reference for the efficient development of low-permeability carbonate rocks.

**Keywords:** low-permeability carbonate rock; fracturing experiments; integrated reservoir model; sweet spots classification; differential stimulation modes

## 1. Introduction

Hydraulic fracturing of the horizontal well is of high cost and risk due to poor physical property and low natural productivity in low permeability and tight reservoirs. In order to control cost and reduce development risk, the concept of geology-engineering integration is gradually widely accepted [1–6]. Its related concepts have been carried out in field practice in tight sandstone, such as the Fuyu oil layer in Daqing Oilfield [7], Yanchang oil group 7 in Changqing Oilfield [8], and Lucaogou oil group in Jimusar Oilfield [9,10]. However, the same stimulation theories and strategies have very different performances in different reservoirs. This suggests that we need to apply targeted stimulation methods and strategies for different reservoirs [11,12]. Even in the same type of reservoirs, such as a carbonate reservoir, due to its diverse sedimentary environments, complex lithology, and microscopic pore throat, the effect of reservoir stimulation is varied and uncertain. Thus, it is critical to carry out related targeted stimulation research [13]. Reservoir characterization and evaluation need continuous improvement based on constantly updated data of experiments, well logging, stimulation, oilfield production, monitoring, and other technical

methods [14,15]. Geology-engineering integration (GEI) method is an effective way to characterize and evaluate unconventional reservoirs [16,17]. Jiang et al. [18] proposed a "double sweet spot" evaluation model to optimize the number and location of fracturing clusters. Xie et al. [19] carried out the integrated design and practice of geoengineering integration in the Changning national shale gas demonstration zone. This paper studies the integrated reservoir model and differential stimulation modes for low permeability carbonate reservoir in X Oilfield, which can make a great reference for the efficient development of low-permeability carbonate rocks.

## 2. Field and Geological Description

The S reservoir is a set of development strata belonging to the X Oilfield. It is vertically divided into subzones, namely SA and SB. SB is an oil-bearing layer, which can be divided into three sublayers: B1, B2, and B3. The average porosity is 16.55%, the average permeability is less than 0.1 mD, and the pore throat radius is less than 0.1 μm. It belongs to low porosity and low permeability carbonate reservoir. There are eight oil wells in production. The natural productivity is weak, with an average daily production of about 200 BOPD. Two vertical wells and one horizontal well have been stimulated in the S reservoir, but the development effect is quite different. The initial daily production of one vertical well after stimulation is 500 BOPD, but there is no stable production period, and it has now been shut in. The initial daily production of the other vertical well and horizontal well after stimulation are 650 BOPD and 1500 BOPD, respectively, and there is a certain stable production period. Therefore, in view of the large difference in production effect after stimulation, it is necessary to formulate targeted development strategies aiming at different reservoir conditions.

## 3. Improved Integrated Strategy of Reservoir Stimulation

Conventional fracturing optimization is mostly based on a one-dimensional rock mechanical model to carry out fracturing fitting of existing wells and optimization design of fracturing parameters [20,21]. This kind of method has the advantages of fast evaluation speed and timely on-site guidance [22,23]. However, at present, with the proportion of low permeability, ultra-low permeability, and tight reservoirs accounting for an increasing proportion, the difficulty and uncertain risk of reservoir stimulation are increasing. The conventional fracturing optimization without experimental guidance can no longer meet the current reservoir requirements. In this paper, the feasibility and post-fracturing conductivity of carbonate rock fracturing are firstly evaluated by conducting laboratory experiments. Secondly, the rock mechanics model, fracturing simulation model, numerical simulation model, and sweet spot model are established so as to clarify the distribution of reservoir stimulation area. Thirdly, the differential stimulation modes for different sweet spot combinations are then proposed, and the fracturing parameters are optimized by analyzing the post-fracturing productivity and net present value. This strategy lays a good foundation for the subsequent overall fracturing design. The specific method of improved reservoir stimulation is shown in Figure 1.

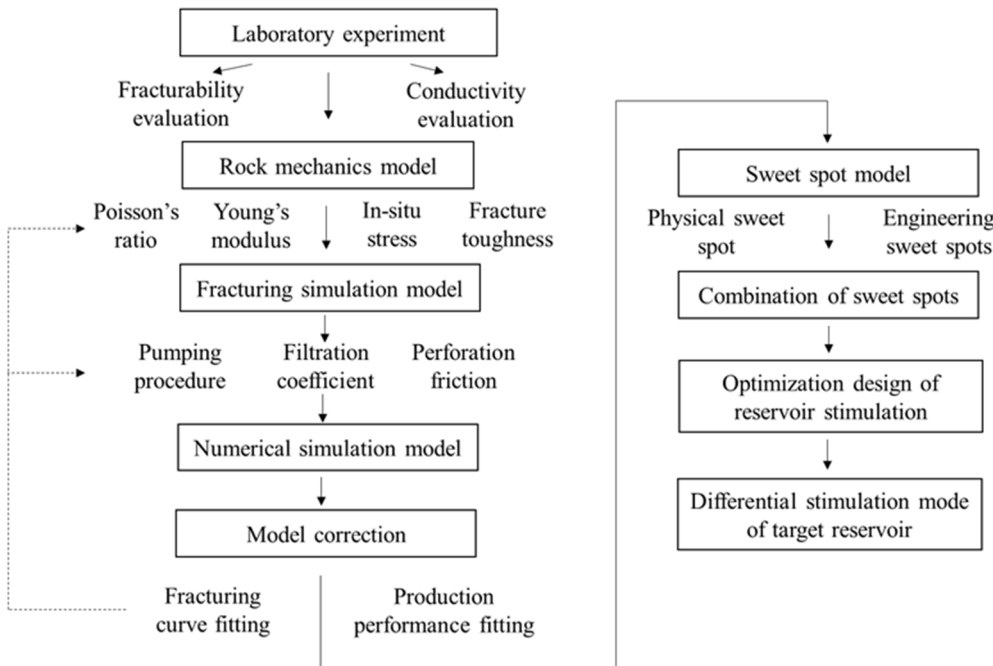

**Figure 1.** The strategy of integrated reservoir model and differential stimulation modes.

## 4. Stimulation Evaluation by Laboratory Experiment

### 4.1. Evaluation of Fracturing Feasibility

Acidizing fracturing is the preferred method for carbonate reservoirs because the rock can chemically react with acid. However, due to the lack of natural fractures in the S reservoir and the low conductivity of the original reservoir, the acid will be largely consumed in the near-wellbore area, resulting in the failure of acid fracturing to give full play to its advantages [24,25]. Therefore, the feasibility of hydraulic fracturing requires relevant experimental evaluation to reduce the reservoir stimulation risk. In this study, the field carbonate outcrop similar to the core property of the S reservoir was selected and processed into massive cores of 300 mm × 300 mm × 300 mm, and casing strings were drilled to simulate casing completion. The properties of the outcrop core and S reservoir core are shown in Table 1.

**Table 1.** Comparison of actual core and outcrop core parameters.

|  | The Average Value of SB2 | The Value of Outcrop Core |
| --- | --- | --- |
| Gas permeability | $0.04 \times 10^{-3} \ \mu m^2$ | $0.028 \times 10^{-3} \ \mu m^2$ |
| Porosity | 18.7% | 15.1% |
| Young's modulus | 14.992 GPa | 13.588 GPa |
| Poisson's ratio | 0.212 | 0.220 |
| Rock mineral analysis | 95% Calcite | 89.1% Calcite |

The slick water was purchased from the Kemaishi Oil Company (product number: DR-800), the experimental concentration was 0.07%, and the viscosity was about 1.25 mPa·s at a shear rate of 170/s. Keeping the injection volume constant, the fracture-forming ability of slick water under different stress conditions was simulated. The experimental parameters and results are shown in Table 2 and Figure 2, respectively. It can be seen from Figure 2 that under different stress conditions, shear fractures can be formed, and the fracture morphology is uniform, which is very beneficial to the fracturing of unconventional reservoirs, indicating that hydraulic fracturing with sand can be carried out in low permeability carbonate rocks.

**Table 2.** Different parameters in fracturing experiments.

| | Fracturing Fluid | Displacement Rate, mL/min | Vertical Stress, MPa | Horizontal Maximum Principal Stress, MPa | Horizontal Minimum Principal Stress, MPa |
|---|---|---|---|---|---|
| 1# | slick water | 60 | 15 | 12 | 2 |
| 2# | slick water | 60 | 20 | 15 | 10 |

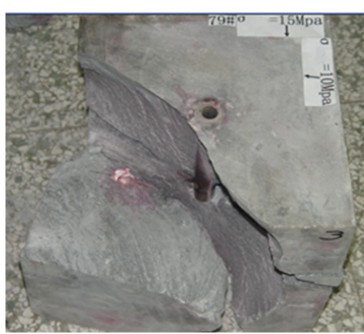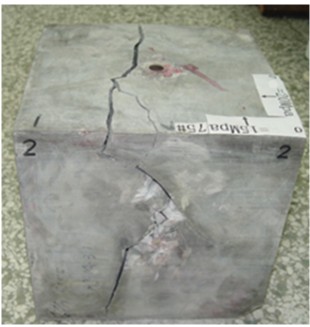

**Figure 2.** Fracturing results in different experimental parameters.

### 4.2. Evaluation of Conductivity

The purpose of hydraulic fracturing is to provide low permeability reservoirs with certain conductivity, while the main factors affecting conductivity are the closure pressure and proppant particle size [26]. Therefore, laboratory experiments are carried out to analyze the influence of proppant particle size and closure pressure on rock conductivity. In this process, sand is firstly laid on the core wall and put into the diversion groove after splitting the actual full-size core. Next, the conductivity is calculated based on the pressure values and then fits the experimental results by the mathematical model so as to realize rapid conductivity evaluation. The mathematical model of conductivity fitting is shown in Equations (1)–(4), and the experimental and fitting results are shown in Figure 3.

$$C_D = Ae^{-0.001B\sigma} \tag{1}$$

$$A = Xd_{\max} - Yd_{\min} \tag{2}$$

$$B = Pm^2 - Qm + R \tag{3}$$

$$m = (d_{\max} + d_{\min})/2 \tag{4}$$

Where $C_D$ denotes conductivity, $\mu m^2 \cdot cm$; $\sigma$ denotes effective closing pressure, Mpa; $d_{\max}$, $d_{\min}$ denote the maximum and minimum particle size of proppant, respectively, $\mu m$; $A$, $B$ are intermediate variables; $X$ and $Y$ are coefficients indicating mesh influence degree, dimensionless; $P$, $Q$, $R$ are coefficients, dimensionless.

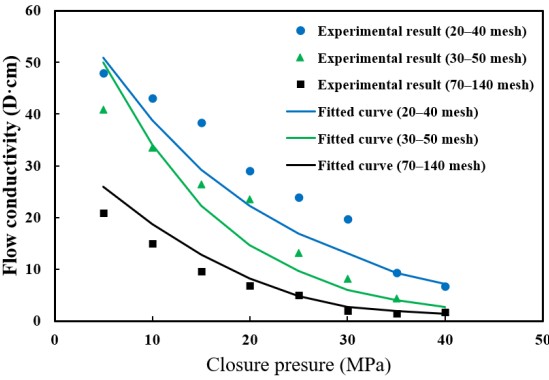

**Figure 3.** Experimental results and fitting results of conductivity under different closing pressures.

## 5. Differential Stimulation Modes Based on Sweet Spot Type

### 5.1. Evaluation of Conductivity

Integrated reservoir models mainly include the rock mechanics model, fracturing simulation model, and numerical simulation model [27]. The establishment of a one-dimensional rock mechanics model is to calculate key rock mechanics parameters such as Young's modulus, Poisson's ratio, maximum and minimum horizontal principal stress, and vertical stress [28]. The calculation formulas are shown in Equations (5)–(9).

$$E_d = 929 \cdot 10^5 \cdot \frac{\rho}{\Delta t_s{}^2} \cdot \frac{3\Delta t_s{}^2 - 4\Delta t_p{}^2}{\Delta t_s{}^2 - \Delta t_p{}^2} \tag{5}$$

$$\mu_d = \frac{0.5\Delta t_s{}^2 - \Delta t_p{}^2}{\Delta t_s{}^2 - \Delta t_p{}^2} \tag{6}$$

$$\sigma_H = \frac{\mu_s}{1 - \mu_s}(\sigma_v - \alpha P_P) + \beta_2(\sigma_v - \alpha P_P) + \alpha P_P \tag{7}$$

$$\sigma_h = \frac{\mu_s}{1 - \mu_s}(\sigma_v - \alpha P_P) + \beta_1(\sigma_v - \alpha P_P) + \alpha P_P \tag{8}$$

$$\sigma_z = 10^6 \int_0^H \rho_r(h)g\mathrm{d}h \tag{9}$$

The calculation results of Equations (5) and (6) are dynamic Young's modulus and Poisson's ratio, but the subsequent fracturing simulation uses static data. Therefore, it is necessary to convert dynamic and static data based on experimental results, namely, regress the dynamic and static experimental data of the same sample or the one that comes from the same depth. In this process, the static data comes from the triaxial stress test results, and the dynamic data is taken from the acoustic characteristics test results, as shown in Figure 4.

For the calculation of maximum and minimum horizontal principal stress, it is necessary to know the horizontal stress tectonic coefficient. According to the fracturing data in the field, the stress coefficients are calculated to be 0.2604 and 0.4154, respectively, based on Equations (7) and (8). Typical rock mechanical parameter curves are shown in Figure 5.

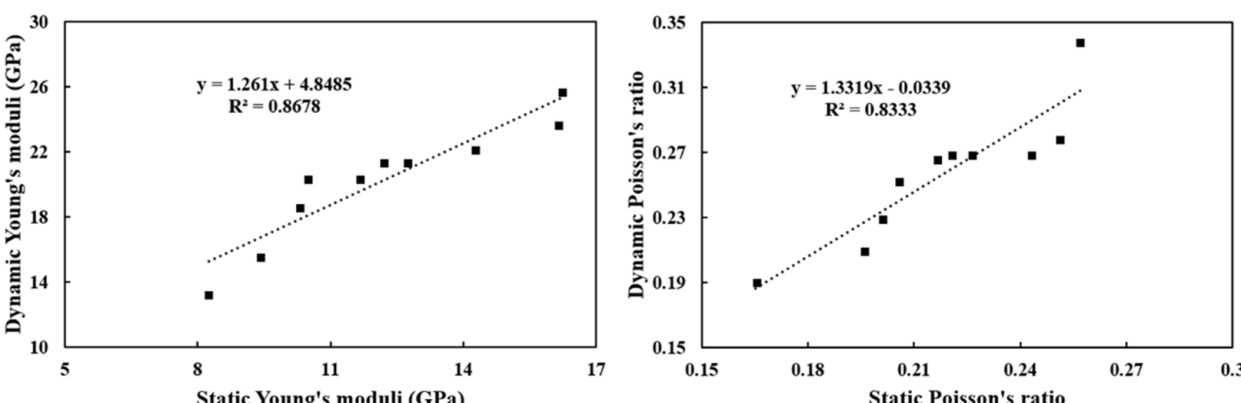

**Figure 4.** Dynamic and static conversion of Poisson's ratio and Young's modulus.

Surrounding rock of overlying and underlying strata and model boundary conditions are constructed. Based on a one-dimensional rock mechanics model, mechanical values are assigned to corresponding rocks. By using rock failure criterion and yield condition, finite element stress-strain simulation is applied to obtain a three-dimensional rock mechanics model of the S reservoir (Figure 6).

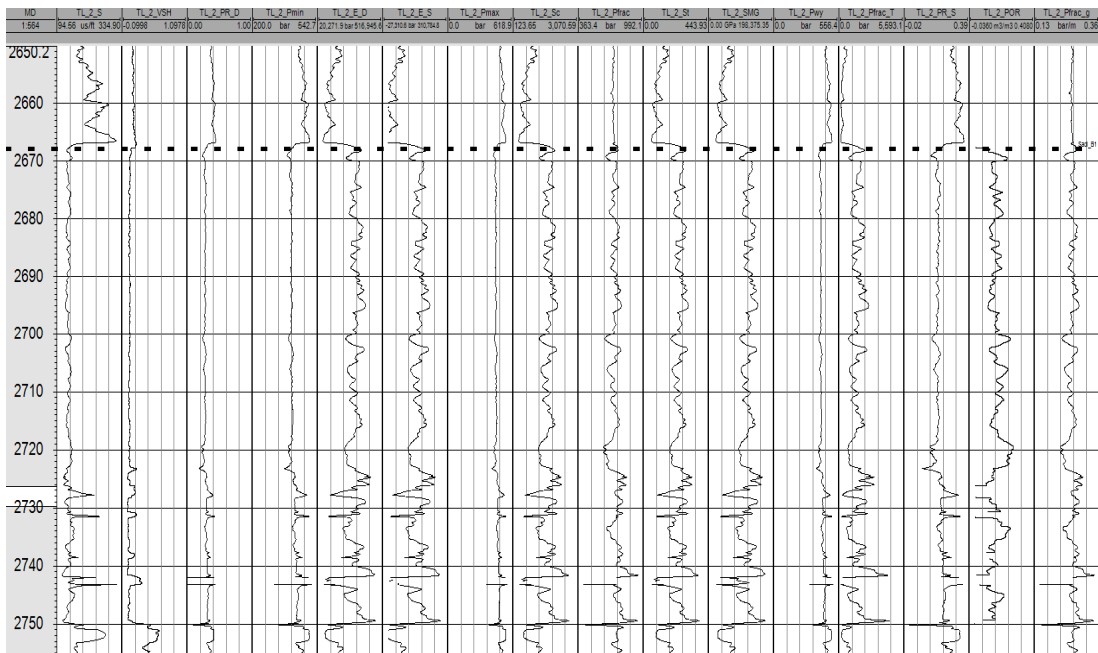

**Figure 5.** Curves of rock mechanics parameters.

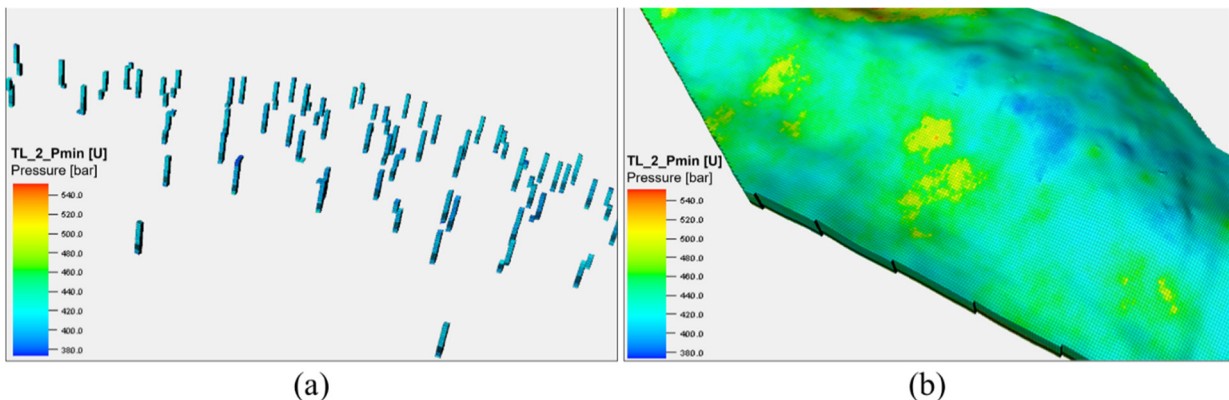

**Figure 6.** Three-dimensional model of minimum horizontal principal stress (**a**) longitudinal dispersion; (**b**) plane prediction.

Based on actual well trajectory, fracturing operation data, the geological model, the rock mechanics model, and boundary element theory, the fracture simulation model is established, which considers the influence of stress shadow on fracture propagation. The fracturing parameters of typical well V5S are shown in Table 3.

**Table 3.** Actual fracturing parameters of typical well V5S.

| Index | Design Parameters | Fracturing Parameters |
|---|---|---|
| Fluid volume ($m^3$) | 426.1 | 463.6 |
| Pre-liquid ratio (%) | 45 | 45.3 |
| Sand amount | 57.2 | 62.0 |
| Average sand ratio (%) | 24.3 | 24.4 |
| Maximum sand ratio (%) | 41.4 | 41.4 |
| Average sand concentration ($kg/m^3$) | 352.9 | 353.8 |
| Maximum sand concentration ($kg/m^3$) | 600 | 600 |
| Pump injection speed ($m^3/min$) | 5.5 | 5–5.5 |
| Mesh number of proppants | 30/50 | 30/50 |

The reservoir numerical simulation model mainly includes unstructured grids, the relative permeability zoning of fracturing and non-fracturing regions, the assignment of fluid high-pressure physical properties, and the formulation of historical or predicted production plans [29]. The purpose of using the unstructured grid is to describe the geometry of the hydraulic fracture network in detail and to calculate the grid permeability according to conductivity and the proppant distribution in the fracture network. The relative permeability zoning and production schedule formulation are consistent with the conventional numerical simulation.

### 5.2. Correction of Integrated Stimulation Model

Due to the uncertain factors of the geological model, rock mechanics model, and fracture simulation model, further correction is needed to ensure the accuracy of simulation results. The correction of the integrated reservoir model is realized by fitting the fracturing operation curve and production performance curve, which aims to correct the main parameters such as rock mechanical parameters, fracturing fluid filtration coefficient, and proppant friction. Partial fitting results are shown in Figures 7 and 8.

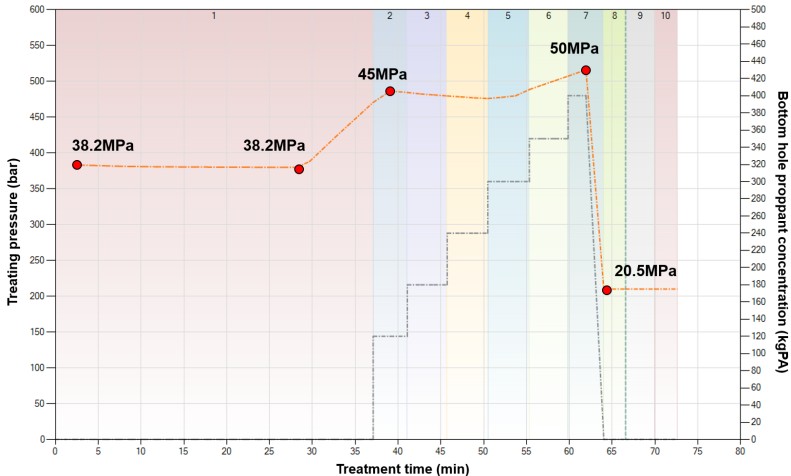

**Figure 7.** The fitting result of fracturing pressure.

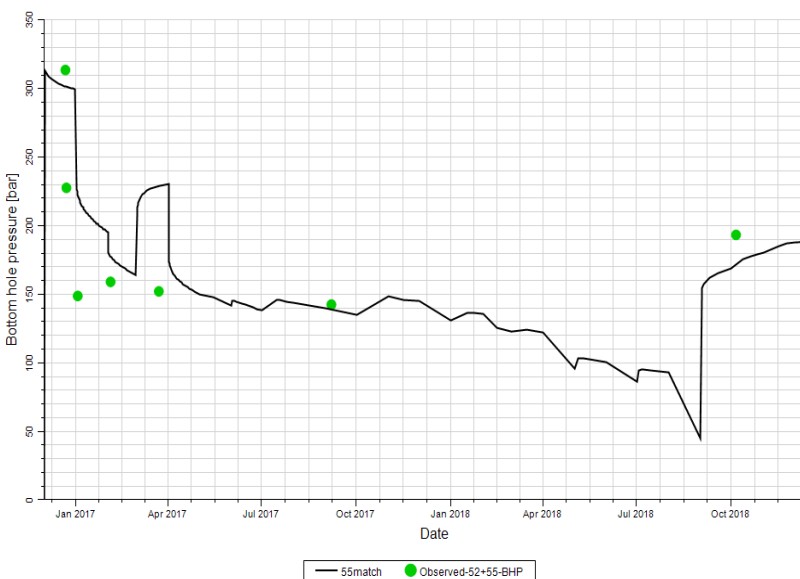

**Figure 8.** The fitting result of bottom hole flow pressure.

The fitting results show that the in situ stress value is about 0.9 times the predicted one, the friction resistance between the proppant and perforation hole is about 1.2–1.5 times the conventional value, the fitting value of fracturing fluid filtration degree is 0.8–0.9 times of actual reservoir, and the Young's modulus of rock is 1.1–1.3 times the predicted one. The other main parameters, such as the shielding layer of the stress profile, Poisson's ratio, vertical stress, fracturing fluid viscosity, and proppant density, are consistent with calculation results and conventional understanding.

### 5.3. Establishment of Sweet Spot Models

The reservoir sweet spot mainly refers to the region which is most favorable to oil production, and is generally considered to be the region with good physical properties and high oil saturation, which is often referred to as the geological sweet spot. The four evaluation indexes of this kind of sweet spot are porosity, permeability, oil saturation, and effective thickness. For low permeability reservoirs, especially reservoirs that need to be stimulated by hydraulic fracturing, engineering sweet spots should also be considered, which characterize the fracturing feasibility. At present, the main three types of rock mechanics parameters that characterize engineering sweet spots are brittleness index, bidirectional stress difference, and fracture toughness [30]. Generally, the larger the brittleness index, the smaller the bi-directional stress difference and the fracture toughness, the higher the fracturing feasibility, and the better the stimulation effect. Comprehensively considering the classification criteria of sweet spots and the actual production of the S reservoir, the classification result of the S reservoir is shown in Tables 4 and 5. The property of sweet spots decreases from A to E. Based on the classification standard, the engineering and geological sweet spot model of the S reservoir are obtained, respectively, as shown in Figure 9.

**Table 4.** Parameter range of different engineering sweet spot.

| Level | Brittleness Index, Decimal | | Minimum Horizontal Stress (MPa) | Fracture Toughness (MPa·m−0.5) |
|---|---|---|---|---|
| | Poisson's Ratio | Young's Modulus (GPa) | | |
| A | >0.9 <0.14 | >20 | <37 | <60 |
| B | 0.8–0.9 0.14–0.18 | 18–20 | 37–40 | 60–100 |
| C | 0.65–0.8 0.18–0.22 | 16–18 | 40–43 | 100–140 |
| D | 0.55–0.65 0.22–0.26 | 14–16 | 43–46 | 140–180 |
| E | <0.55 >0.26 | <14 | >46 | >180 |

**Table 5.** Parameter range of different physical sweet spots.

| Level | Permeability (mD) | Porosity (%) | Oil Saturation (%) | Effective Thickness (m) |
|---|---|---|---|---|
| A | >3 | >23 | >80 | >30 |
| B | 2–3 | 20–23 | 75–80 | 25–30 |
| C | 1–2 | 17–20 | 70–75 | 20–25 |
| D | 0.1–1 | 14–17 | 65–70 | 15–20 |
| E | <0.1 | <14 | <65 | <15 |

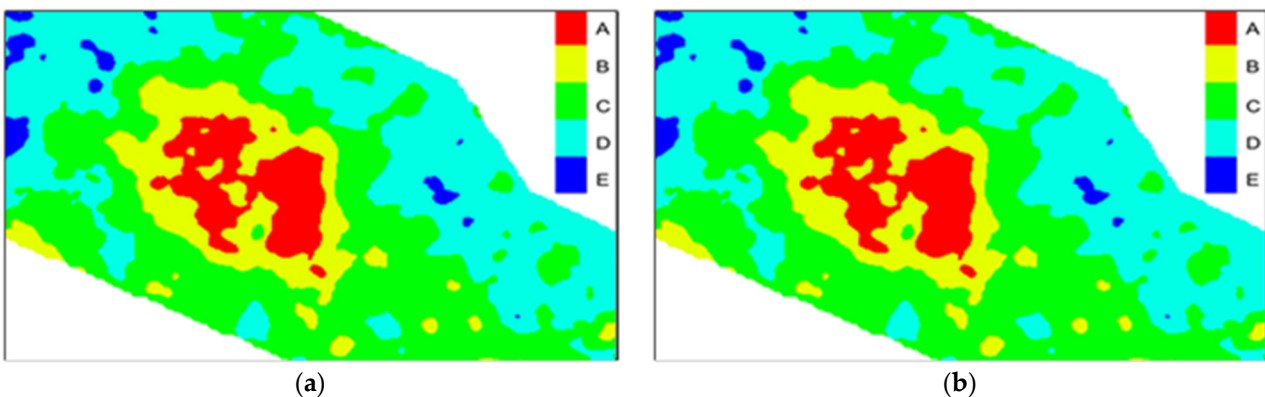

**Figure 9.** Superposition results from different types of (**a**) geological sweet spots and (**b**) engineering sweet spots.

### 5.4. Differential Stimulation Modes

For S reservoir, technical feasibility and economic benefits are taken as objectives in the fracturing design of one horizontal well. In this process, the distribution of both geological sweet spots and engineering sweet spots are taken into consideration, and three types of stimulation areas are divided. By optimizing the fracturing scale of the three, differential simulation modes were formed to obtain the best development performance. Among them, Type I is a combination of geological sweet spot A and engineering sweet spot A. Type II is a combination of geological sweet spot A and non-engineering sweet spot B. And Type III is composed of non- geological sweet spot B and engineering sweet spot A. The numerical simulation results are shown in Figures 10 and 11.

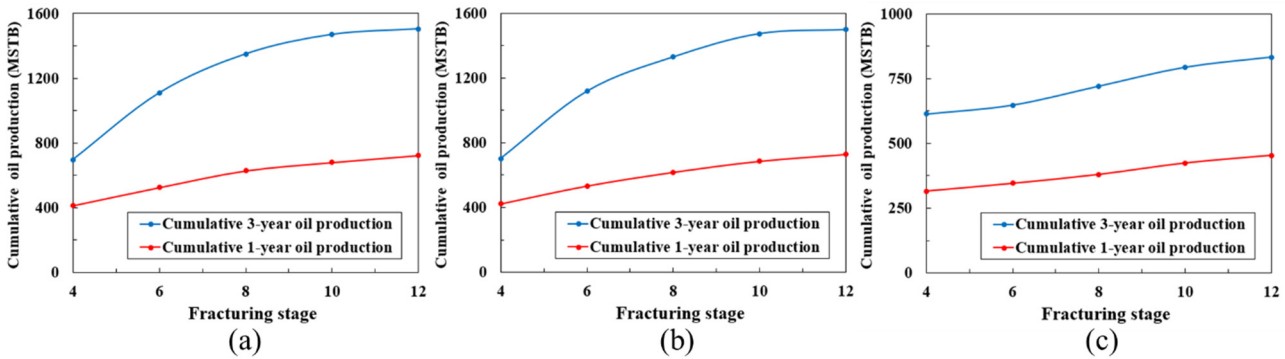

**Figure 10.** Cumulative oil production of stimulation in (**a**) Type I, (**b**) Type II, (**c**) Type III.

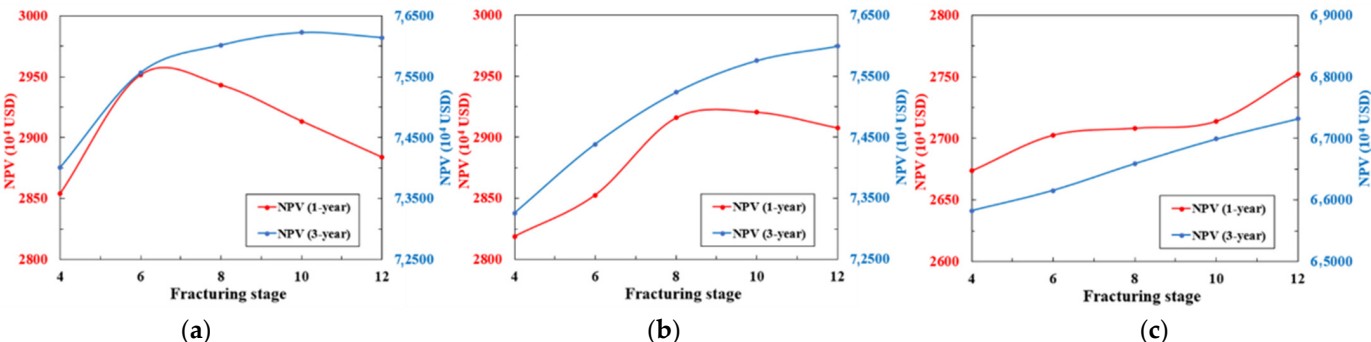

**Figure 11.** NPV of reservoir stimulation in (**a**) Type I, (**b**) Type II, (**c**) Type III.

Figures 10 and 11 show that the cumulative oil production increases with the number of fracturing segments. In Type I, when fracturing segments exceed six, the annual NPV

decreases. Although the NPV is still increasing in the 3-year period, the increasing extent is decreasing. Therefore, considering the economic benefits and post-fracturing productivity, the reasonable number of fracturing segments for the horizontal well is six, that is, the better fracturing effect can be achieved with fewer segments. In Type II, when the number of fracturing segments exceeds eight, the increase of cumulative oil production and economic benefit slows down. Thus, the number of fracturing segments is eight. In Type III, due to the poor physical property, the advantage of engineering the sweet spot should be fully used, namely, producing as many fractures as possible to connect reservoir internal channels. When the number of fracturing segments reaches 12, the increase in cumulative oil production and economic benefit slows down. Therefore, the reasonable number of fracturing segments is 12.

By analyzing the numerical simulation results, differential stimulation modes for different areas are proposed. In Type I, the small-scale stimulation mode can achieve a certain scale of production and maximum economic benefits. In Type II, the medium-scale stimulation mode should be employed to obtain a reasonable production, but the economic benefit is lower than that of Type I. In Type III, the large-scale stimulation mode is required to achieve higher production. This mode is not intended to maximize the economic benefit, but the economic benefit should be above a limit.

## 6. Field Application

A horizontal well H5S with a lateral length of 800 m in Type I was deployed in 2019. According to the optimization results obtained in this study, the reasonable parameter of dimensionless fracture density is one. Thus, the simulation scheme of eight fracturing segments with the same fracturing fluid and proppant was selected. When H5S was put into production, the test output was 1997 BOPD (Figure 12). The initial test production was three times higher than that of fractured vertical wells. The cumulative oil production for one year was about 400 MB. It is expected to obtain economic benefits of USD 242,500 at a USD 60 oil price after deducting the investment.

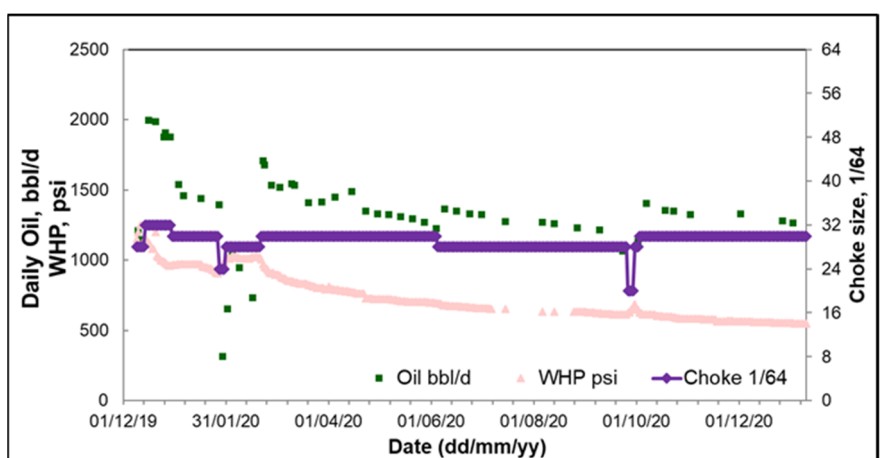

**Figure 12.** Production performance of well H5S.

## 7. Conclusions

Aiming at the problem of the huge difference in the stimulation effect of the low permeability carbonate reservoir in the S reservoir, the feasibility of hydraulic fracturing is demonstrated by laboratory experiments. By establishing the integrated reservoir model, the differential stimulation modes are proposed.

1.  The target reservoir can be stimulated by hydraulic fracturing. The larger the proppant particle size, the better the fracture conductivity after fracturing.

2. Through the fitting of the fracturing operation curve and production dynamic curve, the integrated reservoir model is corrected, and differential stimulation modes are proposed, targeted for the three types of stimulation areas.

3. For Type I, a better economic benefit can be achieved with the small-scale stimulation mode. Type II needs the medium-scale stimulation mode to obtain reasonable productivity, and its economic benefit is lower than that of Type I. Type III requires the large-scale stimulation mode, but the economic benefits should be above the economic limit.

**Author Contributions:** Conceptualization, J.Y. and G.Z.; methodology, J.Y. and G.Z.; experiment, J.Y. and G.Z.; numerical simulation, J.Y. and N.L.; validation, J.Y., G.Z. and W.X.; formal analysis, J.Y. and R.J.; economic evaluation, J.Y., Y.A. and W.X.; investigation, J.Y. and N.L.; resources, N.L. and R.J.; data curation, Y.A.; writing—original draft preparation, J.Y.; writing—review and editing, J.Y. and L.W.; funding acquisition, G.Z. All authors have read and agreed to the published version of the manuscript.

**Funding:** The 14th Five-Year Plan for PetroChina: Research on Key Technologies of Overseas Oilfield Development. Project No.: 2021DJ3203.

**Conflicts of Interest:** The authors declare no conflict of interest.

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
