# Peer review of "Integrated Reservoir Model and Differential Stimulation Modes of Low Permeability Porous Carbonate Reservoir: A Case Study of S Reservoir in X Oilfield in Iraq"

_processes, doi:10.3390/pr10061179_

Round 1
Reviewer 1 Report
Comments to the Author:
I have read the manuscript entitled “Integrated reservoir model and differential stimulation modes of low permeability porous carbonate reservoir: a case study of S reservoir in X Oilfield in Iraq”. An integrated reservoir model and differential stimulation mode are put forward to make a great reference for the efficient development of low-permeability carbonate rocks. The whole paper is well-written and easy to follow. Therefore, I recommend a minor modification before acceptance.
- This paper is innovative to a certain extent, but there are few related researches of previous works. So, it is better to provide some content to the introduction.
- Part 2 could be shortened appropriately.
- Please provide viscosity and manufacturer of fluid used in Part 4.
- Figure 10 is a bit blurry, please provide a clearer picture.
- Please check the reference. Some information is missing, and some papers are duplicated.
- Please check the full text grammar and expression.

Author Response
Dear Associate Editor and Technical Reviewers,
We sincerely appreciate your time and effort on your careful reading this manuscript, understanding all the details, as well as providing valuable comments. These comments were very helpful and we believe have greatly improved the quality of this manuscript.
Author response:
- More details are provided on previous works in the "Inroduction" section.
- Part 2 has been shortened appropriately.
- The viscosity and manufacturer of slick water have been provided.
- More clearer picture of Fig.10 and 11 have been provided.
- The references have been checked.
- The full text grammar and expression have been checked.
Reviewer 2 Report
This paper explores the utilization of hydraulic fracturing in a carbonate reservoir, which is an interesting topic. The authors made tests and simulations, even field trails, which may be valuable. However, the presentation of this paper is more like a project report. For instance, the methods used in this work are regular. Neither novelty nor innovation is detected. Some of the conclusions are common senses. The introduction is too brief and has not fully reported the state-of-art and the literature review. I would encourage the author to redesign the structure of this work, summary and stress the innovation/novelty. It could be a valuable case study by rewriting.
Author Response
Dear Associate Editor and Technical Reviewers,
We sincerely appreciate your time and effort on your careful reading this manuscript, understanding all the details, as well as providing valuable comments. We have incorporated most of the suggestions made by the reviewers and we believe the quality of this manuscript has been greatly improved. The new version of the manuscript will make you and readers of Processes feel satisfied and refreshing.
Reviewer 3 Report
Overall the paper is well written and addresses an important area of research. All the results are presented well, however, Fig. 10 and 11 need to be improved in terms of its visibility.
Author Response
Dear Associate Editor and Technical Reviewers,
We sincerely appreciate your time and effort on your careful reading this manuscript, understanding all the details, as well as providing valuable comments. We have incorporated most of the suggestions made by the reviewers and we believe the quality of this manuscript has been greatly improved. Figures 10 and 11 have been reworked. Thank you.
Round 2
Reviewer 2 Report
/